# AeRChain: An Anonymous and Efficient Redactable Blockchain Scheme Based on Proof-of-Work

**DOI:** 10.3390/e25020270

**Published:** 2023-02-01

**Authors:** Bin Luo, Changlin Yang

**Affiliations:** 1College of Information Science and Technology, Jinan University, Guangzhou 510632, China; 2Pazhou Laboratory, Guangzhou 510330, China; 3School of Software Engineering, Sun Yat-sen University, Zhuhai 519000, China

**Keywords:** redactable blockchain, privacy protection, linkable ring signature, Proof-of-Work

## Abstract

Redactable Blockchain aims to ensure the immutability of the data of most applications and provide authorized mutability for some specific applications, such as for removing illegal content from blockchains. However, the existing Redactable Blockchains lack redacting efficiency and protection of the identity information of voters participating in the redacting consensus. To fill this gap, this paper presents an anonymous and efficient redactable blockchain scheme based on Proof-of-Work (PoW) in the permissionless setting, called “AeRChain”. Specifically, the paper first presents an improved Back’s Linkable Spontaneous Anonymous Group (bLSAG) signatures scheme and uses the improved scheme to hide the identity of blockchain voters. Then, in order to accelerate the achievement of redacting consensus, it introduces a moderate puzzle with variable target values for selecting voters and a voting weight function for assigning different weights to puzzles with different target values. The experimental results show that the present scheme can achieve efficient anonymous redacting consensus with low overhead and reduce communication traffic.

## 1. Introduction

Blockchain can be regarded as a decentralized database and a distributed ledger technology. Compared with traditional databases, blockchain has the important properties of decentralization and immutability. Decentralization effectively solves the problem of a single point of failure, and immutability effectively ensures the integrity of data. These superior properties enable the blockchain to build a trust bridge for nodes in an untrusted environment and ensure the authenticity and reliability of data. Therefore, industry and commerce are considering the use of blockchain technology as a means of solving many practical problems, such as in healthcare [1] and e-government [2].

However, many cases show that immutability limits the development of blockchain. Firstly, some reports mention the existence of illegal data in the blockchain, especially in bitcoin [3]. Secondly, the data stored in the blockchain is increasing, which means that the communication load in the network is constantly increasing. Thirdly, institutions using blockchain technology to provide services sometimes need to correct user data. Because the blockchain cannot be redacted, it becomes very difficult to change wrong data or delete illegal data. Finally, the EU’s General Data Protection Regulation (GDPR) and other regulations require citizens to be given the “Right to be Forgotten” [4], which cannot be met by the current blockchain. Therefore, there is an urgent need for a safe and effective method to redact the data in the blockchain, and Redactable Blockchain is the general term for such methods.

Redactable Blockchain refers to a type of blockchain that allows users to redact the data on the chain in a special scenario. Its redacting operation is mainly aimed at the immutability of the blockchain, that is, to achieve the deletion, modification, and insertion of data on the chain without destroying other properties of the blockchain.

### 1.1. Related Work

In recent years, some scholars have begun to study blockchain redacting functions and put forward some schemes. These schemes can be roughly divided into Chameleon Hash Function (CHF)-based and voting-based redacting consensus methods.

CHF is a one-way hash function with a trapdoor, which can be used to change the input without changing the output of the function. The first redactable blockchain scheme was proposed by Ateniese et al. [5] in 2017. This scheme replaces the original hash function of the blockchain with CHF, which can modify the blockchain data while keeping the hash value unchanged. However, the scheme uses some complex cryptographic technologies which are not suitable for permissionless blockchains. Later, Derler et al. [6] proposed the concept of policy-based chameleon hash. Although the transaction owner can set a policy for redacting the transaction, and only users whose attributes satisfy the policy can redact the transaction, the scheme does not have a mechanism to revoke the trapdoor, so once the user obtains the trapdoor, he may redact data multiple times. Recently, Xu et al. [7] proposed a redactable blockchain scheme with a money punishment mechanism. There is an authority CA in this scheme, which can specify the maximum number of times that redactors can modify the data. Each redactor needs to pay a deposit, and if he acts maliciously, the CA will deduct the deposit as punishment.

In the voting-based method, when the number of votes for the data to be redacted reaches a certain threshold, it can be regarded as having reached a redacting consensus. In 2019, Deuber et al. [8] proposed a redactable blockchain scheme based on this mechanism, which adds an “old state” in the block to maintain the integrity of the hash chain. However, it took too long to reach a redacting consensus in this scheme. To solve this problem, Li et al. [9] proposed an instantly redactable blockchain scheme. In this scheme, a difficult virtual problem is used to run for voters. However, the scheme does not hide the identity of voters, and there is no corresponding voting incentive mechanism. In addition, Thyagarajan et al. [10] proposed a general redactable protocol *Reparo*, which can repair vulnerable smart contracts. However, the protocol may need to cascadingly modify the affected subsequent transactions when necessary.

In order to protect the privacy of users in the redacting process, Cai et al. [11] proposed a removable blockchain that allows users to hide transaction contents and addresses, but this scheme only realizes the deletion operation. Later, Ren et al. [12] proposed a redactable blockchain to ensure the identity privacy of users, but only the transaction sender has the right to modify the transaction. In the same year, Panwar et al. [13] proposed a framework for rewriting blockchain content using CHF and dynamic group signature technology; however, there is a central entity in the scheme.

### 1.2. Our Contributions

Compared with the CHF-based method, the voting-based method can avoid the trapdoor management problem and is more suitable for permissionless settings with a wide range of applications. However, the above privacy protection schemes still have problems such as third-party intervention, centralized redacting rights, and low efficiency in reaching a redacting consensus. Therefore, we have designed an anonymous and efficient redactable blockchain scheme named “AeRChain”, which mainly focuses on how to enable users to participate in the whole redacting consensus process anonymously and compliantly without third-party intervention, and relatively quickly reach redacting consensus in the permissionless setting based on Proof-of-Work (PoW) consensus. The main contributions of this paper are summarized as follows:We improve Back’s Linkable Spontaneous Anonymous Group signatures (bLSAG) scheme, introduce a moderate puzzle with variable target value as a method to select voters participating in redacting consensus, and add a voting weight function to associate the weight of votes with variable target values.We propose an anonymous and efficient redactable blockchain scheme called AeRChain, in which the improved bLSAG scheme is used to hide the identity of voters participating in redacting consensus, and the moderate puzzle and voting weight function are used to accelerate the achievement of redacting consensus.We carry out experiments on the proposed scheme, and the results show that our scheme can achieve efficient and anonymous redacting consensus with acceptable overhead and can reduce communication traffic when the system sets reasonable target values and number of ring members.

The comparison between our scheme and related schemes is shown in Table 1, where public verifiability means that voting results and redacting proofs can be verified by any user.

The rest of this paper is organized as follows: In Section 2, we introduce some basics related to blockchain and cryptography. In Section 3, we show the specific structure of the improved bLSAG scheme and voting weight function, and conduct a security analysis of the improved bLSAG scheme. In Section 4, we first give the system model and threat model of the AeRChain scheme, and then describe the AeRChain scheme in detail. In Section 5, we analyze the security and experimental results of the AeRChain scheme. Finally, Section 6 concludes this paper.

## 2. Background

### 2.1. Blockchain

In this subsection, we refer to the abstract way of describing blockchain in [14,15], review the basic definition of blockchain, and use this abstract way to explain our scheme in the following article. Assume that the blockchain protocol is executed in the discrete time unit *slot*, and that H and G are hash functions in cryptography. In traditional blockchain, a block is of the form $B_{i}:=\left(sl_{i},ne_{i},ph_{i},G\left(tx_{i}\right),tx_{i}\right)$, where $sl_{i}$ is the time unit, $ne_{i}$ is the solution to the underlying consensus algorithm, $ph_{i}$ is the hash value of the previous block, $tx_{i}$ is the transaction contained in block $B_{i}$, and $G(tx_{i})$ is the root of the Merkle tree composed of transactions $tx_{i}$. The Merkle tree is a binary tree obtained by pairwise hashing of transactions. Treat the hash value of each transaction as the leaf node of the tree, and combine two adjacent nodes into a new node from the bottom up until a unique hash value is formed at the end, which is the Merkle root. Blockchain C is composed of a string of the above blocks, i.e., $C:=(B_{1},B_{2},\cdots ,B_{l})$, where $B_{1}$ is the genesis block, $B_{l}$ is the head of the chain, and *l* is the length of the chain. Blockchain mainly uses a consensus algorithm to run for block producers, for example, the PoW algorithm used in Bitcoin. In short, the algorithm takes new block data and a given value *T* as inputs to find a random value *ne* so that H(sl,ne,ph,G(tx))<T.

### 2.2. Linkable Ring Signature

Generally speaking, a linkable ring signature scheme is mainly composed of five algorithms: initialization algorithm *Setup*, key pairs generation algorithm *KeyGen*, signature algorithm *Sign*, signature verification algorithm *Verify*, and link algorithm *Link*.

$Setup(1^{\lambda })$→pp. The algorithm takes the security parameter $1^{\lambda }$ as the input and outputs the common parameter *pp*.KeyGen(pp)→(sk,pk). The algorithm takes the public parameter *pp* as the input and outputs a pair of public and secret keys (*sk,pk*).Sign(R,sk,m)→σ. The algorithm takes ring *R*, signer’s secret key *sk*, and message *m* to be signed as inputs and outputs a signature σ.Verify(σ,R,m)→True/False. The algorithm takes ring *R* and signed message *m* as inputs. If the signature is legal, it outputs True, otherwise it outputs False.$Link(\left(\sigma_{1},R_{1},m_{1}\right),\left(\sigma_{2},R_{2},m_{2}\right))$→True/False. The algorithm takes two sets composed of signatures, rings, and messages as input. If the two signatures are linked, it outputs True, otherwise False.

## 3. The Improved bLSAG Scheme and Voting Weight Function

### 3.1. The Improved bLSAG Scheme

In order to hide the identity of voters participating in the redacting consensus without the intervention of a third party, we consider using the linkable ring signature technology used in Monero [16], namely the Back’s Linkable Spontaneous Anonymous Group (bLSAG) signatures scheme. In theory, it has strong privacy protection characteristics and can provide a secure basis for the blockchain. However, in our redactable blockchain scheme, each voter is allowed to vote for multiple objects in a round, if the original bLSAG scheme is used directly, the voter will be linked, so we add an information identification *id*, and embed it in the linkable tag and signature to allow voters to vote for different objects in the same round. Next, we will give the specific construction after improvement.

Setup. Let *E* represent an elliptic curve defined on a finite field $F_{p}$, *G* be a generator on *E*, and let *d* represent the order of *G*, where *p* and *d* are sufficiently large prime numbers. Further, we need to define the following two hash functions:
$$\begin{array}{c} H_{1}:E\left(F_{p}\right)\to E\left(F_{p}\right).\;H_{2}:{\left\{0,1\right\}}^{*}\to \left\{1,2,\cdots ,d-1\right\}. \end{array}$$The public parameter is $pp=(F_{p},G,E,d,p,H_{1},H_{2})$, and it is used as an implicit input to other algorithms.KeyGen. Each user *u* randomly selects a number $k_{u}$, satisfying $0<k_{u}<d$ as his secret key, and the corresponding public key $K_{u}=k_{u}G$.LRSign. Let $m\in {\left\{0,1\right\}}^{*}$ be the message to sign, and *id* stand for some information identification. Then, the signer chooses some distinct public keys as the ring $R=\{K_{1},K_{2},\cdots ,K_{n}\}$. Let 1≤i≤n represent the signer’s secret index in *R*, and the signer’s public key is $K_{i}$. Signer generates a linkable signature using the following steps:Compute $L=k_{i}H_{2}\left(id\right)H_{1}\left(K_{i}\right)$ as a linkable tag.Select a random number γ∈{1,2,⋯,d−1}, and compute
$$c_{i+1}=H_{2}\left(m,\gamma G,\gamma H_{2}\left(id\right)H_{1}\left(K_{i}\right)\right).$$For j=1,2,⋯,i−1,i+1,⋯,n, pick a random number $r_{j}\in \left\{1,2,\cdots ,d-1\right\}$ and compute $c_{j+1}=H_{2}\left(m,r_{j}G+c_{j}K_{j},r_{j}H_{2}\left(id\right)H_{1}\left(K_{j}\right)+c_{j}L\right)$; where the subscript exceeds *n*, the module *n* is required.Calculate $r_{i}=\gamma -c_{i}k_{i}$(mod *d*).The signature is $\sigma \left(m\right)=\left(R,L,c_{1},r_{1},\cdots ,r_{n}\right)$ with information identification *id*.LRVer. The verifier verifies whether the signature σ(m) is a valid signature created by the signer who holds a private key corresponding to a public key in *R* as follows:Check whether *dL* is equal to 0, if so, then reject.For j=1,2,⋯,n, iteratively calculate
$$c_{j+1}^{\prime }=H_{2}\left(m,r_{j}G+c_{j}K_{j},r_{j}H_{2}\left(id\right)H_{1}\left(K_{j}\right)+c_{j}L\right),$$
where the subscript exceeds *n*, the module *n* is required.Check whether $c_{1}^{\prime }=c_{1}$, if so, accept; otherwise, reject.Link. Given signatures $\sigma^{\prime }=\left(R^{\prime },L^{\prime },c_{1}^{\prime },r_{1}^{\prime },\cdots ,r_{n}^{\prime }\right)$ and $\sigma \prime \prime =\left(R^{\prime \prime },L^{\prime \prime },c_{1}^{\prime \prime },r_{1}^{\prime \prime },\cdots ,r_{n}^{\prime \prime }\right)$, two messages $m^{\prime }$, $m^{\prime \prime }$ and two information identifiers $id^{\prime }$, $id^{\prime \prime }$. The verifier first uses the *LRVer* algorithm to check whether the two signatures are valid. If so, check whether $id^{\prime }=id^{\prime \prime }\wedge L^{\prime }=L^{\prime \prime }$. If it holds, it means that the two signatures are from the same signer for the same information identification, and they will be linked, otherwise the two signatures will not be linked.

### 3.2. Security Analysis

In this section, we will analyze the security of the improved bLSAG scheme. The Elliptic Curve Discrete Logarithm Problem (ECDLP) is a hardness assumption crucial to the security of the scheme, which means that given an elliptic curve *E* defined on a finite field $F_{p}$, a point *G* of order *d* on *E*, and a point *P* that is a multiple of *G*, it is difficult to find an integer a∈[0,d) such that P=aG. Under the hardness assumption of ECDLP, the improved bLSAG scheme is proved to be correct, existentially unforgeable against the adaptive chosen message attack, signer-ambiguous, and linkable through the following theorems.

**Theorem 1** (Correctness). *The improved bLSAG scheme is proved to be correct.*

**Theorem 2** (Existential Unforgeability Against Adaptive Chosen Message Attack). *Under the random oracle model and the ECDLP assumption, the improved bLSAG scheme is existentially unforgeable against adaptive chosen message adversaries.*

**Corollary 1.** *Let $O_{1}\left(R,M\right)$ be a signature oracle that takes the ring set* R*, composed of* n *public keys and the message* M *as input, and generates a signature σ such that LRVer(R,M,σ)=1. An improved bLSAG signature scheme is existentially unforgeable against adaptive chosen plaintext attack if, for any PPT algorithm $A_{1}$ with $O_{1}$ such that $A_{1}^{O_{1}}\left(R\right)\to \left(R,M,\sigma \right)$, for a set* R *of public keys selected by $A_{1}$, and (R,M,σ) is not included in the query-response pair to $O_{1}$, then it satisfies LRVer(R,M,σ)=1 only with negligible probability.*

**Theorem 3** (Signer Ambiguity). *The improved bLSAG scheme is signer ambiguous under the random oracle model and Decisional Diffie–Hellman Problem (DDHP).*

**Corollary 2.** *A improved bLSAG signature scheme is signer ambiguous if, for any PPT algorithm $A_{2}$, inputs of any message* M *, set* R *containing* n *public keys, set of private keys $K=\{sk_{1},sk_{2},\cdots ,sk_{t}\}$, where the public keys corresponding to these private keys belong to* R *, a valid signature σ on (R,M) is generated by user* i *and for any polynomial* P(k)*, where* k *is the security parameter, if $sk_{i}\notin K\wedge 0⩽t<n-1$, we have $Pr\left[A_{2}\left(M,R,K,\sigma \right)\to i\right]\in \left(1/\left(n-t\right)-1/P\left(k\right),1/\left(n-t\right)+1/P\left(k\right)\right)$, otherwise, we have $Pr[A_{2}\left(M,R,K,\sigma \right)\to i]>1-1/P\left(k\right)$.*

**Theorem 4** (Linkability). *The improved bLSAG scheme is linkable.*

**Corollary 3.** *Let $R_{1}$ and $R_{2}$ be two set of* n *public keys. An improved bLSAG signature scheme is linkable if, for all sufficiently large* k*, any $i_{1},i_{2}$∈{1,⋯,n}, messages $M_{1},M_{2}$, information identification $id_{1},id_{2}$, and $\sigma_{1}\leftarrow LRSign\left(sk_{i_{1}},id_{1},R_{1},M_{1}\right)$, $\sigma_{2}$←$LRSign(sk_{i_{2}},id_{2},R_{2},M_{2})$, there exists a PPT algorithm $A_{3}$ which outputs 1 or 0 with $Pr[A_{3}\left(R_{1},R_{2},id_{1},id_{2},M_{1},M_{2},\sigma_{1},\sigma_{2}\right)$$=1\wedge \left(i_{1}\neq i_{2}\vee id_{1}\neq id_{2}\right)]⩽\xi \left(k\right)$ and $Pr[A_{3}\left(R_{1},R_{2},id_{1},id_{2},M_{1},M_{2},\sigma_{1},\sigma_{2}\right)$=0∧$i_{1}$$=i_{2}\wedge id_{1}=id_{2}]$⩽ξ(k), where ξ is a negligible function.*

The complete proofs of the above four theorems are in the Appendix A, where the proof of Theorem 1 is in Section A.1, the proof of Theorem 2 is in Section A.2, the proof of Theorem 3 is in Section A.3, and the proof of Theorem 4 is in Section A.4.

### 3.3. Voting Weight Function

In order to achieve flexible and efficient redacting consensus, we introduce a voting weight function. Each user who wants to participate in the redacting consensus obtains different voting weights by solving puzzles with different target values. The smaller the target value, the greater the difficulty and the greater the weight, which will enable the whole network to reach redacting consensus faster.

**Definition 1** (Voting Weight Function F). *A voting weight function F is a mapping from target value to weight, and its form is as follows, where* v *is the maximum voting weight determined by the system and i∈N.*
$$\begin{array}{c} F\left(T_{i}\right)=k_{i}.\;T_{i}\in \left\{1,2,2^{2},\cdots ,2^{256}\right\},k_{i}\in \left\{1,2,\cdots ,v\right\} \end{array}$$

The setting of the voting weight function needs to be combined with the system parameters and constraints, and both it and the target value of the moderate puzzle to select voters are mainly related to the target value *T*′ of the underlying consensus mechanism of the blockchain system. Generally speaking, the target value of the voting weight function is a multiple of *T*′, and the corresponding weight is some small values. For example, the system can set $F\left(T_{1}\right)=1,F\left(T_{2}\right)=3,T_{1}=116T^{\prime },T_{2}=78T^{\prime }$, refer to the parameter settings in Section 5.

## 4. The Proposed Anonymous and Efficient Redactable Blockchain Scheme

### 4.1. System Model

The system diagram of our AeRChain scheme is shown in Figure 1. The roles of the AeRChain scheme mainly include users, block producers, and voters. Note that the division of each role is not independent, for example, a user may be a block producer and a voter at the same time.

Users. Users are also called ordinary users in the blockchain system and are the main members of the blockchain system. Users can be the sender or receiver of the transaction, they can also become block producers through the underlying consensus mechanism, voters participating in the redacting consensus, or other users unrelated to the transaction.Block Producers. Block producers, also called miners, are the main force for maintaining the blockchain system. They are responsible for collecting transactions and packaging them into new blocks to expand the blockchain. In our AeRChain scheme, block producers are also responsible for collecting and aggregating votes, updating the status value of redacted blocks, and replacing old blocks with new redacted blocks.Voters. Voters are selected from users through a moderate puzzle called custom Proof-of-Work (cPoW), and obtain corresponding voting weights according to the voting weight function. Voters are mainly responsible for anonymous voting on redacting requests sent by users, and are core members of the redacting consensus process.

The main idea of the scheme is that if a user *j* wants to redact the content of a block $B_{i}$, the user can make a redacting request, including the index *i* and the redacted content $B_{i}^{*}$. Other users can vote for the redacting request by solving moderate puzzles cPoW and obtaining voting weights *w* according to the voting weight function. After that, the voters anonymously vote on the redacting requests within a specified time. When users receive a vote, they verify whether the voting information is correct, and whether the voters are legal. If the verification is passed, the weight value contained in these voting data will be accumulated. When the value reaches a certain threshold *ts*, it indicates that the whole network has reached a consensus on the redacting request. Then, the block producer aggregates the corresponding proofs, packs them together with ordinary transactions into a new block, and adds the block to the blockchain. At the same time, the block producer is also responsible for updating the content of redacted blocks. Finally, he broadcasts the new chain $C^{\prime }$, then other users update the local chain after receiving and verifying $C^{\prime }$.

### 4.2. Threat Model

In our AeRChain scheme, the random algorithm cPoW is used to select voters. Its principle is similar to the underlying PoW consensus, which can ensure that the selected users are the honest majority, but there may still be malicious users among them, so the voters are semi-trusted. Similarly, the miners elected through the underlying PoW consensus are also the honest majority, so they are also semi-trusted. Ordinary users are untrustworthy, and they may jointly launch collusion attacks with malicious voters or malicious miners. In summary, there may be the following threats in the AeRChain scheme: (1) Ordinary users vote for redacting requests and try to pass the voting verification. (2) Malicious voters vote repeatedly or abuse voting rights in an anonymous environment, that is, they vote for the same object multiple times in the same round, or vote in the same round with more weight than they themselves have, and attempt to escape the linkability of anonymous signatures. (3) Malicious voters vote for maliciously redacted transactions and try to get the voting weight to reach the threshold. (4) Other users try to generate anonymous signatures of legal voters, and link pseudo-anonymous signatures with real voter signatures. (5) In the case of a voter performing a legitimate vote, other users attempt to obtain the voter’s identity information in order to bribe the voter.

### 4.3. Detailed Description of the AeRChain Scheme

In order to add a redactable function to the blockchain, we have slightly expanded the block header of the traditional blockchain. Specifically, we add a list variable *oh* in the redactable block to store the historical information of the Merkle Tree root of the block. Because the root value $G(tx_{i})$ of the Merkle tree in the traditional block is implicit in *oh*, we remove it so that the block is now in the form of $B_{i}:=\left(sl_{i},ne_{i},ph_{i},oh_{i},tx_{i}\right)$. If the block has not been redacted, then $oh_{i}$ is equal to $G(tx_{i})$. Otherwise, $oh_{i}$ is a list that stores all the historical Merkle tree root values of the redacted block, i.e., $oh_{i}=\left(oh_{i}^{0},oh_{i}^{1},\cdots ,oh_{i}^{c}\right)$, and *c* is the number of times the block has been redacted, $oh_{i}^{0}$ is the initial Merkle tree root of the block, and $oh_{i}^{c}$ is the current Merkle tree root value of the block. For convenience of explanation, we assume a block can be redacted at most once, but it is easy to expand to multiple times by changing *oh* to a list.

In addition, the AeRChain scheme uses rounds to promote the execution process of the blockchain. If the voting period is *q* slots and the network delay is Δ slots, then one round is q+Δ slots. The specific duration of each round is independent of the block generation time and is determined by the network environment and system parameters.
Initialisation:

When the blockchain system is initialized, the system generates the public parameter *pp* through the *Setup* algorithm and broadcasts it to all users. Each user obtains their public key $pk_{i}$ and private key $sk_{i}$ through the *KeyGen* algorithm, and then makes $pk_{i}$ public, so that any user knows the public keys of other users in the system. When the blockchain is initialized, blockchain C has only one genesis block, which contains some necessary information related to the system, including public parameters, system parameters, and user public keys. All users maintain a local blockchain C and a list *EditVote* that collects voting information, which is updated regularly.
Generating redacting requests:

When a user wants to redact block $B_{i}$, the user can generate and broadcast the corresponding redacting request, which includs the index *i* of the block to be redacted and the corresponding redacted content $B_{i}^{*}$ (i.e., candidate block). In addition, each user who requests redacting needs to pay a redacting fee, which is determined by different systems, such as according to the scope and quantity of redacting content. When the whole network reaches a consensus on a redacting request, the redacting fee will be distributed according to the voting weight of voters. Since voters are anonymous, voters can submit relevant proof information of voting and a one-time address for allocating awards after reaching a consensus.
Redacting consensus:

Redacting consensus is the core of all redacting steps, including generating voters, voting for candidate blocks, and updating the blockchain. To prevent adversaries from voting multiple times for the same redacting request in the same voting period, we added a key binding identification, so that if an adversary uses multiple pairs of keys to do this, these votes will not pass the verification. If user *i* wants to participate in an anonymous redacting consensus, *i* first selects a ring member set *R* composed of users’ public keys, and then uses the appropriate target value *tv* set by the system to run for voters by solving moderate puzzles we call “custom Proof-of-Work” (cPoW), and vote on the redacting request during the current voting period *q*.

Suppose the current blockchain is C, the blockchain head is $B_{l}:=\left(sl_{l},ph_{l},oh_{l},ne_{l},tx_{l}\right)$, and the new block is B=(sl,ph,mt,tx), where *mt* is the Merkle tree root G(tx) of the new block. The *cPoW* Algorithm 1 takes the new block header bh:=(ph,mt,sl), the target value *tv* of the *cPoW*, the voting weight function F, the key binding identification $lid:=k_{i}H_{1}\left(K_{i}\right)$, and the ring member set *R* as inputs, where *ph* is the hash value of $B_{l}$, *sl* is the start timestamp of the current round, and $K_{i}$ and $k_{i}$ are the public and secret keys of the user calling the algorithm, respectively. Then, the user looks for some random value *ne*, so that the hash value of these data is smaller than *tv*. After successfully finding *ne*, the voter will obtain the voting weight *w* and the corresponding proof *info*, so in this round, he will become a voter and can vote anonymously on the redacting request within the current voting period *q* slots.

**Algorithm 1** Custom Proof-of-Work Algorithm *cPoW*
**Input:** 
the new block header bh:=(ph,mt,sl), the target value *tv*, the voting weight function F, the key binding identification *lid*, and the ring member set *R***Output:** 
weight *w* and proof *info*   1:parse w:=0, info:=⌀, the current timestamp is tm;   2:**while** 
tm<sl+q+Δ
**do**   3:  **if** find *ne* to make H(tm,ph,ne,mt,H(lid||R))<tv
**then**   4:    w=w+F(tv), info=info∪(ph,tm,ne,mt,tv,lid,R);   5:**return**(w,info);


The voter generates a voting message *m*, which is composed of information identification *id*, ring member set *R*, voting weight *w*, and proof information *info*, where *id* includes the hash value of a candidate block and the serial number *r* of the current round. Then, the voter uses the algorithm *LRSign* to sign *m* and broadcasts the voting information vm(m,σ). When other users receive votes from the network, they verify them using the *VerifycPoW*, *LRVer*, and *Link* algorithms, and add the verified *vm* to *EditVote*. The *VerifycPoW* Algorithm 2 takes the proof information *(w,info)*, the voting weight function F, linkable tag *L* used in improved bLSAG, and information identification *id* as the input. The verification process is similar to *cPoW* algorithm, except that the judgment on the correctness of the voter’s identity information is added. 

**Algorithm 2** Verifying cPoW Algorithm *VerifycPoW*
**Input:** 
weight *w*, proof *info*, the voting weight function F, linkable tag *L*, and information identification *id***Output:** 
0 or 1   1:parse info:=(ph,tm,ne,mt,tv,lid,R), c:=0, the hash value of the chain head at tm−1 slot is $ph^{\prime }$, the start timestamp of round *r* is sl;   2:**for** each *info*
**do**   3:   **if**
$H(tm,ph,ne,mt,H(lid||R))<tv\wedge ph=ph^{\prime }\wedge L=H_{2}\left(id\right)lid\wedge tm\in \left[sl,sl+q+\Delta \right)$
**then**   4:    c=c+F(tv);   5:
**if**

c=w

**then**
   6:   **return** 1;   7:**return** 0;


When the cumulative weight of votes for a candidate block reaches a certain threshold *ts*, the block producer aggregates all the proof information contained in the votes into a redacting proof, packs it together with ordinary transactions into a new block $B^{\prime }$, then links it to the blockchain head. At the same time, the block producer replaces the original block $B_{i}$ with the redacted block $B_{i}^{*}$, and attaches the Merkle tree root of the old block $B_{i}$ to the state variable *oh* in the block header of $B_{i}^{*}$. Finally, the new chain $C^{\prime }$ is broadcast. After other users receive it, the algorithm *LRVer*, *VerifycPoW*, and *Link* are used to verify the legitimacy of $C^{\prime }$ and update the local blockchain.

Different systems have different thresholds, which need to exceed the maximum number of votes that malicious users can generate. Because we mainly focus on the process of redacting consensus, the steps for users to verify the validity of candidate blocks and the whole chain after reaching a redacting consensus can be referred to [8]. Note that redacting consensus and ordinary block consensus are parallel, that is to say, users may also become voters in the process of running for block producers, but it depends on whether users want to participate in redacting consensus.

## 5. Security Analysis and Experimental Evaluation

### 5.1. Security Analysis

In this section, we analyze the security of the proposed AeRChain scheme, which is mainly based on the hardness of ECDLP and the collision-resistant property of the hash function.

Firstly, the AeRChain scheme with anonymity function satisfies the following security properties:Signer anonymity. Voters use the improved bLSAG scheme to sign their votes, because everyone in the ring has an equal position and can generate a signature; unless the real signer exposes himself, the verifier cannot identify who is the signer. Therefore, this property ensures the identity privacy of voters.Linkability. In our redactable blockchain scheme, the linkable tag is composed of the signer’s secret key, the public key, and the voting information identification. If each user has a unique key pair, because the serial number of the voting round is also unique, the linkable tag must be unique. If the user holds multiple key pairs, the key binding identification associated with one of the key pairs needs to be input before calculating the *cPoW* solution, so the key information cannot be changed after obtaining the solution. In other words, each pair of keys corresponds to a puzzle. Only when the puzzle is successfully solved can the corresponding voting weight be obtained. If the adversary wants to increase the voting weight, a certain number of puzzles must be solved. Therefore, when a voter votes for the same candidate block in the same round with more than the weight he obtains, these votes will be linked and cannot pass the legitimacy verification, so this property prevents repeated voting in the redacting consensus.Unforgeability. On the one hand, the security of the improved bLSAG signature scheme is based on the hardness of ECDLP, which means that for any PPT algorithm A, the probability that Pr[A(G,aG)=a] is negligible, where $G,aG\in E(F_{p})$. If a voter wants to forge the signature, he needs to obtain the signer’s secret key by solving the ECDLP. On the other hand, since the solution of each *cPoW* is bound to a key pair, if the voter wants to change the key pair after finding the solution, he needs to find a hash collision that satisfies the solution. However, as we all know, it is difficult to solve the ECDLP and find a hash collision, and no one can forge or replace the signature except with negligible probability.

Secondly, because only users who successfully solve the *cPoW* will get the right to vote, even if they vote anonymously, the solution of the *cPoW* puzzle is bound to the information of the ring set and key binding identification used for the anonymous signature. Therefore, the votes of ordinary users who fail the election will be rejected by verifiers, meaning that the AeRChain scheme can resist threats (1).

Thirdly, since the key binding identification binds each *cPoW* solution to a pair of key pairs, once the malicious voters with a pair of keys repeatedly vote, their votes will not pass the verification of the *Link* algorithm. For malicious voters with multiple pairs of keys, they can only obtain more voting weight by solving multiple cPoWs. Otherwise, once the key is replaced and the vote is repeated, their vote will not pass the *Link* algorithm, *LRVer* algorithm, and *VerifycPoW* algorithm verification. Thus, by embedding the key binding identification and ring member set into the puzzle of selecting voters, no matter how many pairs of keys a voter has, in an anonymous environment the verifier will reject multiple votes by the same voter for the same object in the same round, or reject votes whose weight do not match the voter’s vote weight. This means that the AeRChain scheme can defend against threats (2) and (3). In addition, the AeRChain scheme is resistant to threats (4) and (5) due to its signer anonymity, linkability, and unforgeability shown above.

In summary, based on the security analysis of the improved bLSAG scheme (Section 3.2), and through the combined verification of key binding identification, *VerifycPoW*, *Link*, and *LRVer*, the AeRChain scheme can defend against all threats listed in the threat model (Section 4.2).

Finally, the anonymous and efficient redactable blockchain C satisfies the characteristics of chain quality and chain growth, such as ordinary immutable blockchain $C^{\prime }$, and follows the definition of [14]. Because C is essentially an extension of $C^{\prime }$, it does not remove blocks from the chain. The *cPoW* algorithm of our scheme is parallel to the *PoW* algorithm of traditional blockchain, that is, the solution of *cPoW* may also be obtained during the process of running *PoW*, which will not affect the basic block consensus or slow down the original chain growth. Therefore, if the immutable blockchain $C^{\prime }$ meets the chain growth property, then C will also meet the chain growth property.

In addition, due to the collision-resistant property of the hash function, C also satisfies the property of chain quality and the extended definition of common prefix given by [9], i.e., the redactable common prefix, which is suitable for blockchains with a redaction function. The redaction operation is introduced in our scheme, and adversaries may modify the honest block B to the malicious block $B^{*}$. However, when an adversary makes a redacting request that includes candidate block $B^{*}$, since the adversary only accounts for μ(μ<1/2) parts of the total computing power, it can be seen from the “no long block withholding” lemma [17] that, as long as the system sets a reasonable voting weight threshold *ts* and a reasonable voting weight function F, the probability of the adversary successfully redacting block B is negligible unless honest users vote for $B^{*}$. In addition, if an adversary wants to construct a candidate block of $H\left(B^{*}\right)=H\left(B\right)$ to cheat honest users to vote for it, then, due to the collision-resistant property of the hash function, the probability that the adversary can succeed is negligible, where $B^{*}\neq B$, $B^{*}$ is malicious content, and B is normal content. Therefore, if $C^{\prime }$ satisfies the chain quality property, then C will satisfy the chain quality property.

The redactable common prefix means that if the chain $C_{1}$ with the length of $l_{1}$ and the chain $C_{2}$ with the length of $l_{2}$ are owned by two honest users $u_{1}$ and $u_{2}$ at slot $sl_{1}$ and $sl_{2}$, respectively, where $sl_{1}⩽sl_{2}$. One of the following two conditions must be satisfied: (1) The prefixes of $C_{1}$ and $C_{2}$ consisting of the first $l_{1}-k$ records are identical, where k is a parameter of the common prefix. (2) For each $B_{i}^{*}$ in the prefix of $C_{2}$ consisting of the first $l_{1}-k$ records, but not in the prefix of $C_{1}$ consisting of the first $l_{1}-k$ records, it must mean that the whole network has reached a redacting consensus on $B_{i}^{*}$ at the time of $sl_{i}\;\left(sl_{1}<sl_{i}<sl_{2}\right)$, and its proof is stored in the first $l_{1}-k$ blocks.

### 5.2. Parameters Settings

The parameter settings in our scheme refer to [9], and the following constraints and relationships need to be satisfied:

We assume that the number of users in the whole network is *n*, the proportion of adversaries is a(a<1/2), and the voting period is *t* slots. The adversary finds the solutions of the *cPoW* puzzle *q* slots earlier than honest users. Let $q\approx k*2^{l}/T^{\prime }$, where *k* is the common prefix parameter, *l* is the output length of the hash function and $T^{\prime }$ is the target value for the underlying PoW blockchain. Let $\alpha_{i}$ and $\beta_{i}$ represent the number of cPoWs expected to be solved by honest users and adversaries, respectively, in each slot when they select the target value $T_{i}$. Accordingly, $N_{h}$ and $N_{a}$ represent the maximum number of *cPoWs* solved by honest users and adversaries in *t* and t+q slots respectively. We take $N_{a}$ as the voting threshold *ts*.

For any δ>0, with negligible probability $P_{1}=exp^{-(\delta *min\left\{\delta ,1\right\}*\left(\sum_{i=1}^{max}k_{i}*\beta_{i}\right)*\left(q+t\right))/3}$, it holds that $N_{a}>\left(1+\delta \right)*\left(\sum_{i=1}^{max}k_{i}*\beta_{i}\right)*\left(q+t\right)$, where $k_{i}$ is the voting weight corresponding to $T_{i}$. For any δ∈(0,1), with negligible probability $P_{2}=exp^{-(\delta^{2}*\left(\sum_{i=1}^{max}k_{i}*\alpha_{i}\right)*t)/2}$, it holds that $N_{h}⩽\left(1-\delta \right)*\left(\sum_{i=1}^{max}k_{i}*\alpha_{i}\right)*t$, because we need legitimate voters who can solve cPoW puzzles to be an honest majority, so we need to guarantee $N_{h}>N_{a}$. Then, we assume that $T_{i}=x_{i}*T_{max}$, 1⩽i<max, $k_{j}=y_{j}*k_{1}$, 1<j⩽max, where $x_{i}$ and $y_{j}$ are two intermediate variables, $x_{i}$ represents the proportion between each target value and the minimum target value, and $y_{j}$ represents the proportion between each voting weight and the maximum voting weight. The relevant parameters are substituted into $N_{h}>N_{a}$ and reduced to the following formula:$$\begin{array}{cc} t & >q/(\left(\left(1-a\right)\left(1-\delta \right)\left(x_{1}h_{1}+\sum_{i=2}^{max-1}x_{i}y_{i}h_{i}+y_{max}h_{max}\right)\right)\\ \; & /(a\left(1+\delta \right)\left(x_{1}a_{1}+\sum_{i=2}^{max-1}x_{i}y_{i}a_{i}+y_{max}a_{max}\right))-1). \end{array}$$

According to the above constraints, and for generality, we finally set these parameters as follows: a=0.4, voting weight function $F\left(T_{1}\right)=1,F\left(T_{2}\right)=3$, δ=0.1, and set $h_{1}=0.2,h_{2}=0.8,a_{1}=0.6,a_{2}=0.4$, then t>1.7q, so we set q=10,t=18. If $P_{1}=exp^{-15},P_{2}=exp^{-14}$, then $T_{1}=116T^{\prime },T_{2}=78T^{\prime }$, and the voting threshold *ts* is approximately 4950, this value is the cumulative weight of different votes, so the total number of votes is less than this value, and these votes are cast during the voting period of 18 slots.

### 5.3. Experimental Evaluation

We used Python language to construct a simplified blockchain system based on PoW consensus, which simulates some basic functions of Bitcoin. Then, in order to realize the redactable function of the blockchain based on voting consensus, we added a list variable *oh* in the block header to store the old Merkle root value of the block. Our experimental environment is an Ubuntu 16.04 (64bits) system, and the relevant configuration is AMD Ryzen 5 3600 6-Core Processor with 3.6 GHz and 8 GB of RAM. Next, we conducted a series of experiments and evaluations based on the constructed blockchain. Because our scheme utilizes the improved bLSAG scheme and cPoW algorithm, the difference in experimental results is mainly related to the number of ring members and the target values.

First, we evaluated the average time required to verify a vote, as shown in Figure 2a. It can be seen from the figure that even if the number of ring members reaches 50, the time for verifying votes is acceptable. For example, a ring with 11 members is currently recommended in Monero. Therefore, the cost of verifying votes is negligible relative to the whole redacting consensus process.

Secondly, we evaluated the average size of a vote, as shown in Figure 2b. It can be seen from the figure that even though there are many ring members, the size of the vote is small relative to the network communication load.

Thirdly, as shown in Figure 2c, we evaluated the time required for our scheme to reach a redacting consensus under the different number of ring members when each block contains a different number of transactions, and compared the results with scheme [9]. In the figure, we use “Li” to represent [9] scheme, “RN” represents the number of ring members and shows the time required to reach a redacting consensus when the number of ring members is two, four, six, and eleven. On the one hand, the time of our scheme is far less than the 513 slots of [8]. On the other hand, in our scheme, under the average case, as long as each vote contains no more than six ring members, the time to reach a redacting consensus is faster than Li et al. When the number of ring members exceeds seven, taking the number of eleven ring members currently recommended by Monero as an example, our scheme is at most two slots slower than Li et al. Therefore, when the number of ring members is reasonable, compared with the existing voting-based efficient redactable permissionless blockchain scheme, our scheme can reach anonymous redacting consensus at a faster speed.

Finally, we evaluated the size of the redactable blockchain in the average case according to the different percentages of the deleted block data in the whole chain. As shown in Figure 2d, we fixed the length of the chain to 1000 blocks, where each block contains 3000 transactions, and set the number of ring members to eleven as required by the current Monero. The size of an ordinary immutable blockchain is about 0.96 GB. After using AeRChain, with the deletion of more and more block data, the size of the redactable blockchain will be much smaller than that of an ordinary blockchain, which can reduce the communication traffic in the network.

As redacting data in the blockchain is an important operation, redacting operations should only be allowed under special circumstances, and data cannot be redacted frequently. Therefore, an efficient redacting scheme should specify the maximum number of times that data can be redacted within a period, and at the same time make the time of reaching a redacting consensus within the user’s acceptable range, such as a few hours. Although the improved bLSAG scheme seems to bring some overhead compared with ordinary signatures, compared with the permissioned blockchain, there are usually only a few redacting requirements in the permissionless blockchain. In our scheme, the overhead is mainly determined by the number of ring members. Therefore, as long as the system specifies or recommends a reasonable number of ring members to users according to the actual application scenario, the redacting consensus is efficient and the overhead is acceptable.

## 6. Conclusions

In this paper, we propose an anonymous and efficient redactable blockchain scheme based on PoW in a permissionless setting, called “AeRChain”. Firstly, we improve the bLSAG scheme and use it to hide the identity of voters participating in the redacting consensus. Secondly, we introduce a voting weight function and a moderate puzzle with variable target values to improve the efficiency of redacting consensus. Finally, the experimental results show that, compared with existing schemes, our scheme can achieve efficient and anonymous redacting consensus with a low overhead, and can reduce communication traffic.

## Figures and Tables

**Figure 1 entropy-25-00270-f001:**
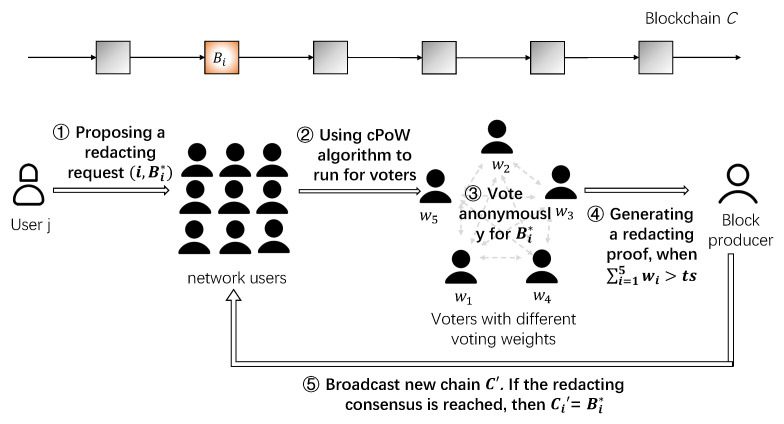
AeRChain system diagram.

**Figure 2 entropy-25-00270-f002:**
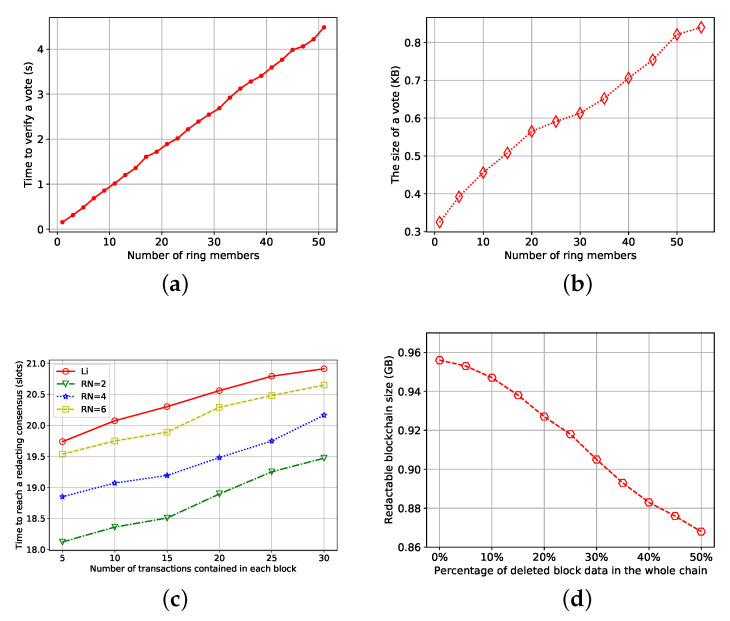
Experiment of AeRChain scheme. (**a**) Time to verify a vote; (**b**) The size of a vote; (**c**) Time to reach a redacting consensus; (**d**) Redactable blockchain size.

**Table 1 entropy-25-00270-t001:** Comparison of our scheme with related works.

Scheme	Permissionless	Public Verifiability	Redaction Time	Anonymity Without Third Party
Deuber [8]	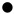	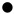	513 slots	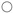
Li [9]	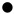	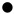	20 slots	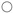
Panwar [13]	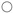	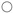	—	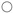
ours	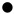	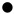	≤ 20 slots	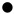

## Data Availability

The data presented in this study are available on request from the corresponding author.

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
