# Peer review of "AeRChain: An Anonymous and Efficient Redactable Blockchain Scheme Based on Proof-of-Work"

_entropy, 2023, doi:10.3390/e25020270_

Round 1

Reviewer 1 Report

1.     The voting weight function in Section 3.3 needs to be explained in detail, including the purpose and role of using it, and how to set it to make the voting weight function reasonable.

2.     In the parameter settings in Section 5.2, it is necessary to clearly explain the meaning of the parameters ki, xi and yi, and how to calculate these parameters.

3.     The current experimental results do not show the comparison with existing schemes, so these comparison figures can be added. For example, the number of ring members can be fixed, and the proposed scheme can be compared with the existing scheme.

4.     Some statements are lengthy and need to be simplified. For example, “the above privacy protection schemes ... and cannot balance the privacy and efficiency well”.

5.     In the security analysis in Section 5.1, the analysis of chain quality, chain growth and redactable common prefix is relatively simple. It is suggested to expand the analysis of this part appropriately to make the security analysis more detailed.

6.     Some grammatical mistakes in the paper weaken the readability.

Reviewer 2 Report

1.     In the security analysis of the improved bLSAG scheme, the proof of the "Signer Ambiguity" property is simple and needs further improvement.

2.     If the security analysis of the improved bLSAG scheme occupies more pages, it can be moved to the appendix.

3.     The introduction of voting weight needs to be more detailed and explain what kind of voting weight function is reasonable and how to set it.

4.     In parameter settings, the interpretation of parameters is somewhat vague, and the meaning of parameters needs to be clearly stated. In addition, it is necessary to further clarify whether there are some constraints between the set parameters and the system parameters.

5.     The security analysis in Section 5.1 needs to be more detailed.

6.     There are some grammar and sentence mistakes in the paper, such as: “Assume that the blockchain protocol is executed in discrete time unit slot, and let H and G be hash functions in cryptography...”.

Round 2

Reviewer 1 Report

I have no further comments, and I think the current version is good for publication. 
